# Students' experience of interpersonal interactions quality in e-Learning: A qualitative research

Rita Mojtahedzadeh[1], Shirin Hasanvand[2,3]*, Aeen Mohammadi[1], Sahar Malmir[4], Mehdi Vatankhah[5]

1 Department of E-Learning in Medical Education, Center of Excellence for E-learning in Medical Education, School of Medicine, Tehran University of Medical Sciences, Tehran, Iran, 2 Social Determinants of Health Research Center, Nursing and Midwifery School, Lorestan University of Medical Sciences, Khorramabad, Iran, 3 E-Learning in Medical Education, Tehran University of Medical Sciences, Tehran, Iran, 4 Nursing and Midwifery School, Lorestan University of Medical Sciences, Khorramabad, Iran, 5 Medical Education Development, Lorestan University of Medical Sciences, Khorramabad, Iran

* hasanvand.sh1390@gmail.com.

**Data Availability Statement:** All the relevant data files are available from the figshare database 'https://doi.org/10.6084/m9.figshare.23599155.v2'

## Abstract

### Background

Online Interaction is a critical characteristic of distance learning, and effective online communication models empower students.

### Purpose

This research aimed to explain students' experiences on the quality of interpersonal interactions in e-learning.

### Method

This study was conducted from November 2021 to October 2022. The qualitative descriptive design via conventional content analysis was utilized. Purposeful and maximum variation methods recruited sixteen participants from three medical science universities in Iran. The data were collected through semi-structured, in-depth, face-to-face, or online interviews. Interviews were recorded through a digital recorder, and analysis was achieved simultaneously with data collection using Graneheim and Lundman (2004). The Lincoln and Guba criteria, including credibility, dependability, transferability, and confirmability, were used to improve the trustworthiness of the findings.

### Results

The results indicated the importance of different dimensions related to teaching-learning. It seems crucial to develop a comfortable and safe environment to improve interpersonal interactions. Educators should be provided with pedagogical skills to support interactions. In addition, focusing on some learners' soft skills is also vital. In addition to the significance of the teacher's inclusive role, the educational content must have critical standards.

**Funding:** The author(s) received no specific funding for this work.

**Competing interests:** The authors have declared that no competing interests exist.

**Abbreviations:** LMS, Learning Management Systems.

Constructive feedback and the proper use of simultaneous and non-simultaneous communication tools and social networks are other important issues in strengthening interpersonal relationships. Ultimately, comprehensive and ongoing support of learners improves the quality of interpersonal interactions.

## Conclusions

The results indicated the significance of different dimensions of teaching-learning as facilitating factors of interpersonal interactions. The proper use of simultaneous and non-simultaneous communication tools and social networks are other important issues in strengthening interpersonal relationships. Ultimately, comprehensive and ongoing support of learners improves the quality of interpersonal interactions.

## Implications

The results of this study give teachers the insight to keep essential issues in mind when developing their online courses and students to be aware of their roles in the online learning process. Also, the characteristics of simultaneous and non-synchronous platforms, social messaging networks, and learner support are crucial.

## Background

Interaction is one of the common topics of research in e-learning, and it is one of the basic elements of any learning experience [1, 2]. Moreover, interaction is a critical element of the pedagogical process whereby teachers, students, and the learning content share a common learning environment. There are three main pedagogical interaction types: learner-to-content, learner-to-instructor, and learner-to-learner [3].

Online interaction is one of the essential elements of distance learning [4]. Being knowledgeable of communication weaknesses in online environments can help educators to develop appropriate materials [2]. Given the role of interaction between the teacher and the learner and, and given the importance of meeting the needs of different students, in distance teaching, teachers should find new ways to communicate with their students and use new communication methods appropriate for synchronous and asynchronous learning environments [5].

However, students' interaction experiences in online environments differ from face to face (F2F) classes [6]. The main feature of online education is the physical distance between the teaching staff, peers, and educational institutions. Therefore, different technological tools should be used to fill this gap to improve interactions [4].

The teacher-student interaction during the teaching-learning process, and specifically in distance education, can affect students' satisfaction, motivation, and ability to manage learning tasks [5].

However, it is important to note that high levels of interaction do not necessarily facilitate effective learning. Mehall et al., indicate that interaction should be structured, purposeful, and accompanied by leadership, and that interaction quality is more meaningful than measuring its quantity [1].

## Literature review

### Interpersonal communication

Communication is the transfer of information from one person or group to another person or group, and it is a process for exchanging thoughts, knowledge, and information in a way that realizes the goal or intention in the most suitable possible way [2]. In the most conventional conceptualization, communication is the creation of significant messages that are sent or received from one person to another or between people or groups [4].

Ghazi Saidi et al.(2021) introduced the fundamental teaching characteristics based on the student's perception. Accordingly, there were five characteristics: careful attention, respect, motivation, giving feedback, and providing lectures. According to their findings, the students' interpersonal relationships were essential [7]. Interaction can include the interaction of the instructor with the learner, the learners interacting with each other, and the learner with the content [8].

Learner-content interaction is the capability to access and interact with information/content [9]. Interaction with students is one of the most important responsibilities of teachers, both in traditional curriculum and in online courses. The ease of communication influences the interaction between the instructor and the student, the degree of students' comfort in raising questions and expressing their ideas, and how to access the instructor and respond to the problems related to information [6]. Instructor-learner communication refers to establishing a comprehensive connection with instructors and receiving feedback from them. Interpersonal communication between the students and the instructors mainly emerges in the class during lectures. When the teacher gives a lecture and the students listen, sending and receiving messages occur, resulting in sufficient communication. Effective communication is undoubtedly dependent not only on messages but also on specific characteristics in which interpersonal communication can be effective [9]. Learner-learner interaction includes the interaction of learners with each other and the creation of a learning community. Moore highlights three types of interaction, and his transactional distance theory is derived from these three types of interaction [10]. This article addresses the latter two kinds of interaction in e-Learning.

### E-Learning

Learning delivered via information and communication technologies is usually regarded as e-learning [11]. E-learning is sometimes referred to as online learning, web-based learning, computerized learning, computerized teaching, and other equivalent words [12]. The concept of e-learning is very wide. It was forged in the late 90s as a technologically enriched learning instrument via the Internet. Currently, it catches a wide range of electronic media like the Internet, Intranets, and Extranets to make the learning procedure more adaptable. Because of the flexible character of e-learning, it has more demand and requirements among the people of the world than ever before [13].

Compared to traditional settings, e-learning is more self-pace [14]. The findings of Mastour et al. in Iran revealed that students' learning results were more significantly pronounced e-teaching than in the traditional in-person teaching. They concluded that e-learning could be relished as a successful mode of medical teaching, and that it might be utilized as an alternative educational method [15]. Among the advantages of e-learning, one can mention the following: teaching regardless of time and place, cost-effectiveness, no need for physical presence, the existence of various choices, time-saving, and the development of cooperative learning types [16].

Despite growing evidence that e-learning is as efficacious as traditional methods(Maulana et al., 2023), it arrives with challenges. It was determined in the study of Ahmed et al. 2023 that

the challenges encountered by students were a poor internet connection and a lack of interest and desire [12]. The need for complex technology, reduction of social interactions, unstable internet connection, inability to understand students' facial expressions, and the impossibility of performing practical and laboratory courses are among the weaknesses of distance education [16].

## Quality of interpersonal interaction in E-learning

It is specifically critical to understand communication processes in online communities [17], and similar to the goals of face-to-face communication, the goal of online interaction is exchanging information, being heard, and being understood [2]. Moreover, online interactions are one of the basic elements of distance learning, and the use of effective online communication models is very important for empowering distance education learners and for preventing negative experiences such as isolation and loneliness during the course of learning [4]. Online communication enables students to use messages, images, audios, and videos of asynchronous or synchronous interaction systems. Asynchronous communication does not happen in real-time. On the contrary, simultaneous communication takes place in real time and is facilitated by learning management systems (LMS) [4].

In e-learning, as in many other fully online courses, researchers are concerned about challenges that may hinder student learning, including the lack of human interaction in virtual learning. In particular, learning in a virtual learning environment is especially significant due to the physical separation between students and instructors [18]. Physical separation innately causes a psychological and communication gap between the instructor and the learner, and therefore, it might lead to misinterpretation between the teacher and the learners [19].

In two studies, they mentioned the challenges raised in relation to online learning. The challenges included the instructor's inability to teach the material virtually, not receiving appropriate feedback from students, lack of academic coherence such as the lack of discipline and the possibility of student cheating, doubts about the accuracy and validity of information, concerns about network security and about the privacy of users. Moreover, one might add other challenges and/or requirements: The need for a proper foundation for the proper use of virtual networks and the existence of a strict monitoring system to check and organize the information available in virtual networks [20, 21]. An online survey found that students in online classes face many problems, such as decreased motivation and comprehension, significantly decreased communication between them and the teachers, and isolation cases [2]. This is specifically true for Iran's higher education since there are prominent challenges related to e-learning in this country in terms of interactions. One study enumerated the main challenges in terms of educational, organizational, ethical, technical, supportive and managerial aspects, and particularly, in terms of evaluation and communication [22]. Sason et al. (2022) reported that during the COVID-19 pandemic, students insisted that they needed to interact with their teachers. From the learners' perspective, the most common form of interaction was e-mail-mediated educational Interaction. The minor form of interaction was social intimacy. In their qualitative study, Goudarzi et al. (2023) introduced the category called the challenges of the teaching-learning process [16].

Many studies have confirmed the relationship between learners' engagement and interpersonal interactions [23–25]. Other studies have also emphasized learner-teacher and learner-learner interactions in academic achievement, learning, and satisfaction [24, 25].

The quality of interpersonal interactions is a neglected link in e-learning. Hunter and Ross (2019) study confirmed a positive linear relationship between interactions for each student and the quality perceived by a student in a course. Their study showed that increasing interaction alone could improve students' experiences and perceptions of quality [8]. The experiences

of the respondents of another study indicated the importance of both quality and quantity of interactions in online learning [10]. To our knowledge, therefore, no study has focused on the quality of interpersonal interactions in e-learning.

Educators often seek to replicate the dialogue that is easily achievable in their F2F courses in the online setting by utilizing discussion forums and similar techniques. However, educators have not come to consensus concerning the best interpersonal interaction strategies for an effective student learning / satisfaction. Teachers are frequently pressured to improve the quality of their online courses. However, they have to familiarize themselves with different strategies to encourage students to interact. In other cases, college teachers have been teaching in an F2F environment for years. They are being asked to convert their courses into an online format without pedagogical and technical support [1, 26].

Additionally, many studies on interaction in the online environment do not consider the qualitative aspects of interaction, and they only measure the number of interactions, which typically occurs through such methods as measuring discussion forum posts or course updates [1, 26]. Social and instructional interactions among students and student-instructors are common elements of an F2F classroom. In the F2F classroom, many interactions happen spontaneously. F2F learning provides many opportunities for informal learning where interaction is not planned. In the online environment, this informal learning will take place if students are given the opportunity and appropriate tools to interact with their peers and instructors. For this reason, quality instructional and social interaction opportunities in online environments need to be deliberately designed into the courses [1, 27].

Unfortunately, few studies deal with the quality of interpersonal interactions. In most studies, the experiences of learners have been considered in terms of e-learning effectiveness [28, 29], and some studies have placed emphasis on the learners' perspective on e-learning challenges [22]. To our knowledge, no study focuses on background and contextual distinctions of learners when reflecting on the quality of interpersonal interactions. For this reason, the present study tries to explain students' experiences of the quality of interpersonal interactions in e-learning.

## Methods

### Design

This study was conducted from November 2021 to October 2022. The qualitative descriptive design via conventional content analysis was used to explore students' experiences of interpersonal interaction quality in e-learning. Content analysis is a technique for creating replicable and reasonable abstractions from texts or other significant topics to the contexts of their use [30]. This type of design is usually appropriate when the existing theory or research literature about a phenomenon needs to be revised. Researchers do not use predetermined categories. Instead, it allows categories and category labels to flow from the data [31].

### Participants and data collection

The participants were recruited from three medical science universities in Iran. Students of Tehran University of Medical Sciences (TUMS) and Shahid Beheshti Universities of Medical Sciences (SBMS) were selected from two virtual faculties. A group of students were from the Tehran University of Medical Sciences Virtual School. Tehran Virtual School was established for the first time in 2012. It takes students with master's degrees in e-learning in Medical Sciences, Educational Technology and doctoral degrees in e-Learning by the blended method. Faculty of Medical Education and Learning Technologies of Shahid Beheshti University of Medical Sciences has been taking Medical Education and e-Learning students since 2013. Students study

in a blended learning. The Lorestan University of Medical Sciences students were from various disciplines of Medical Sciences who had online learning experience during the COVID-19 era.

Deep and semi-structured interviews were conducted with 16 Persian-speaking students. The primary criteria for including the participants were willingness to participate in research, share experiences, and experience with online teaching for at least one academic semester. The participants were selected by purposeful and maximum variation in gender, age, marital status, semester, educational level, and field of study.

The first participant was a faculty member who was studying as a student in e-learning in Medical Sciences and as a teacher and manager during the COVID-19 era. The participant had the experience of online teaching and addressing problems related to e-learning. In the same way, based on each interview and the obtained data, the next participant to be interviewed was selected.

Before data collection, the participants received a phone illustration of the project. Informed consent was sent the day before the interview, online or in person, and after obtaining informed consent, the participants were interviewed.

The data were collected through semi-structured, in-depth, F2F, or online interviews. Semi-structured interviews, primarily used in health care, include several key questions that help define the research area. In semi-structured interviews, the interviewer utilizes open-ended questions "to debate some subjects in more detail." By open-ended questions, the interviewer gathers corresponding data from the interviewee(s) with personal ideas [32].

Firstly, the interview guide was compiled based on a literature review and previous experience talking to experts. The order of questions was different for all participants. As the interviews continued and new aspects of the interviewees' experiences emerged, questions were modified, and other items were added. An interview guide was used (Table 1).

The first interview was treated as a pilot interview to confirm that the interview questions were relevant and appropriate for the purpose. The interview lasted 20 to 60 minutes(40 minutes on average). Interviews were recorded through a digital recorder and conducted by one of the researchers (corresponding author) with adequate qualitative research experience. The interview continued until data saturation. Data saturation is reached when the interviewer concludes that no new themes or explanations emerge from the next interviews.

## Data analysis

One of the researchers (the corresponding researcher) performed the interviews and manually transcribed the audio recordings. The interviews were analyzed in the manifest form, the meaning of the participants' descriptions of their experiences, and the latent form, the

**Table 1. Interview guide.**

| Participants' experiences regarding teacher-student interactions |
|---|
| 1- Tell me about your experience interacting with the teacher in the virtual class. |
| 2- How do you see interaction in synchronous online classes? How about asynchronous? |
| 3- According to your experience, what factors affect the quality of teacher-student interactions? |
| 4- If you have had a good or bad experience with a synchronous or asynchronous online class, can you tell me what the characteristics of such a class were? |
| 5- Tell me about your experience interacting with peers in the virtual class. |
| 6- How do you see interaction with peers in synchronous online classes? How about asynchronous? |
| 7- In your opinion, what factors affect the quality of student-student interactions? |
| 8- If you have had a good or bad experience with a synchronous or asynchronous online class, can you tell me what the characteristics of such a class were? |

underlying meanings of the participants' expressions. Analysis was accomplished simultaneously with data collection and using Graneheim and Lundman (2004) (33)with the following stages:

1. Immediate transcription of interviews

2. Listening to them to obtain a general perception

3. Identification of significant components and initial codes(the label of a meaning unit)

4. Category of similar initial codes in more comprehensive categories (creating categories)

Immediately after the interview, the text of the interview was transcribed, then typed and reread several times so that the researcher could get a general sense of the interview. Then, the rereading of the interviews was continued to determine the meaningful parts. We assume a meaning unit as words, sentences, or paragraphs, including dimensions about each other via their content and context. The condensed meaning units were outlined and labeled by a code. Afterward, the codes were organized using a constant comparison method, recognizing differences and similarities and identifying subcategories. Finally, the findings were compared, and categories and themes were identified.

## Trustworthiness

The results should be as authentic as possible, and each study must be assessed regarding the procedures used to create the finding [33]. The Lincoln and Guba criteria, including credibility, dependability, transferability, and confirmability, were used to increase the trustworthiness of the findings. To increase credibility, we choose participants with various experiences of the phenomenon under study, with various genders and ages, educational levels, marital statuses, member checking, and fields of study. The long-term engagement of the researcher (12 months) was also considered. In order to determine dependability, the research team participated in the study process, and the findings were presented to several external observers to review the data analysis process. A clear description of the student's culture, context, and characteristics and the data collection and analysis process were provided to increase data transferability. The confirmability was accomplished by bracketing, reporting, and documenting the study phases and decisions accurately so that other researchers could track it if preferred and the audit trail could be conducted on analysis.

## Ethics approval and consent to participate

This research was done under the Declaration of Helsinki. The ethics code was obtained from the Deputy for Research and Technology of Tehran University of Medical Sciences (IR.TUMS. MEDICINE.REC.1400.686). After providing explanations about the study's objectives, written informed consent was obtained from all participants. One of the crucial challenges of electronic and online interviews is ethical considerations and getting consent to participate. For this purpose, the consent is sent one day before the interview through email or social networks for participation so that they can read it. The interview is done after obtaining the consent, signing it, and sending the consent form to the researcher.

## Results

This study was conducted on 16 students with a mean age of 33.9 years. 55% of students were women. Most of the participants (67%) were married. Among the participants, 2 were students and faculty members simultaneously, and the rest were students. Most students were graduate

students, and only 2 had undergraduate education. Table 2 displays an overview of the demographic data of participants. Data analysis showed three themes and eight categories. The main themes include teaching and learning factors, features of the platforms, and systemic factors. The results of the interview are summarized in Table 3.

## Theme 1. Teaching and learning factors

Learners' experiences showed that different aspects of teaching and learning, including instructor characteristics, learners, content, and subject matter, can be practical in interpersonal communication in e-learning.

Most of the students highlighted creating a positive and motivational environment for learning. A friendly atmosphere with trust, reducing tension and anxiety, a comfortable and dynamic environment without discrimination and prejudice, and being an energetic teacher were among the characteristics that the students noticed. The teacher's pedagogical skills, such as academic capability, a good presentation technique, ability to class management, making an environment for active participation, improving learners' engagement, paying attention to the learning style of the learners, and getting feedback from the students were also a critical issue in the interaction between the teacher and the learners. Also, the learners stressed having charisma and the essential knowledge and skills in software and applicable instructional design.

If the teacher leaves the student-teacher relationship in the classroom and can be comfortable, I think this is the best interaction to let this student-teacher relationship turn into a friendship" (P.10).

**Table 2. Characteristics of participants.**

| Participants (No) | Gender | Age (year) | Previous study | Marital status | University | Field of study | Semester | Degree |
|---|---|---|---|---|---|---|---|---|
| **P1** | Male | 20 | Midwifery | Single | LUMS[*] | Nursing | 5 | Bachelor |
| **P2** | Male | 45 | France language | Married | TUMS [**] | e-Learning | 4 | MSc |
| **P3** | Male | 20 | - | Single | LUMS | Midwifery | 6 | Bachelor |
| **P4** | Male | 46 | Medical Records | Married | TUMS | e-Learning | 3 | MSc |
| **P5** | Male | 50 | Immunology | Married | TUMS | e-Learning | 4 | MSc |
| **P6** | Male | 30 | Health Information Technology | Single | SBMU[***] | Medical Education Technology | 6 | MSc |
| **P7** | Male | 46 | Medical Education | Married | SBMU | e-Learning | 2 | MSc |
| **P8** | Male | 22 | Dentistry | Single | SBMU | e-Learning | 2 | MSc |
| **P9** | Male | 35 | Medical Records | Married | TUMS | e-Learning | 4 | MSc |
| **P10** | Male | 33 | Nursing | Single | TUMS | Medical Education Technology | 4 | MSc |
| **P11** | Male | 30 | Health Information Technology | Married | TUMS | e-Learning | 4 | MSc |
| **P12** | Female | 29 | Health Education and Health Promotion | Married | SUMS | e-Learning | 6 | MSc |
| **P13** | Male | 40 | Environmental Health Engineering | Married | SBMU | e-Learning | 6 | MSc |
| **P14** | Male | 40 | Radiology | Married | SBMU | e-Learning | 8 | MSc |
| **P15** | Male | 28 | Medical Education | Single | TUMS | e-Learning | 4 | PhD |
| **P16** | Male | 29 | Medical Education | Married | TUMS | e-Learning | 4 | PhD |

* Lorestan University of Medical Sciences

** Tehran University of Medical Sciences

*** Shahid Beheshti University of Medical Sciences

**Table 3. The theme, main categories, and subcategories obtained from content analysis.**

| Theme | Main categories | Subcategories |
|---|---|---|
| **Teaching and learnings' Factors** | Instructor characteristics | Creating a positive learning environment |
| | | Having pedagogical skills |
| | | Having charm and charisma |
| | | Having ICT knowledge and skills |
| | | Effective instructional design |
| | | Provide appropriate feedback to learners. |
| | | Appropriate assignment design |
| | Learner characteristics | Homogeneity of learners |
| | | Having soft skills |
| | | being busy |
| | | Having knowledge and skills in the field of ICT |
| | | the number of learners |
| | Content and subject characteristics | Electronic content design according to standards |
| | | Logical sequence of lesson presentation |
| | | Applicability of the subject |
| | | previous familiarity with the subject |
| **Features of the platforms** | Asynchronous platforms | LMS user-friendliness |
| | | Permanent access to content in the LMS |
| | | Ability to track overall performance and progress in LMS |
| | | Proper navigation of the discussion forums |
| | | Absence of LMS technical problems |
| | Synchronous platforms | The possibility of sharing screen |
| | | Interruption and frequent system connection |
| | | The possibility of grouping learners in small classes |
| | | Providing the possibility to hear the voice and see the image and body language of the teacher |
| | Social Networks | Facilitate and strengthen interaction with social networks. |
| | | Informing social networks |
| **Systemic factors** | Learner support | Educational support |
| | | Technical support |
| | Extracurricular activities | Providing the possibility of F2F interactions |
| | | Nonacademic activities |

The atmosphere of the class was much more sincere. We did not feel anxious and worried in that class, and I think the interaction was effortless" (P.2).

A teacher needs to manage a class well " (P.10).

In online teaching, teachers use different communication tools " (P.6).

All participants emphasized the importance of receiving positive feedback based on principles, transparency, and respect, aiming to improve performance.

" *Now that I have entered the* learning management systems *(LMS) in my sixth semester, some of my assignments related to the previous semesters have no feedback, even though it has been a long time, and I have no hope anymore* " *(P6).*

" Feedback should be respectful " (p16).

" The teacher's feedback method is critical. How clear it is, for example, some professors write that you said no wrong " (p9).

Discussion forums are asynchronous communication tools widely used in learning management systems that require the teacher's active participation. According to students, teachers should engage students with these discussion forums and follow up permanently, not temporarily. Some learners emphasized the help of learners in managing the discussions and encouraging students to give fair criticism.

" Unfortunately, there were few chat rooms with few interactions. It was because the teacher's presence was less " (P6).

" Forums are excellent environments. It can be said that the most place where we could understand students' engagement with virtual education was in these forums " (P4).

" The teacher can get help from other people to manage the discussion forum " (P15).

" What I think is very effective in the interaction is that students must comment on each other's comments in the discussion forum. Criticize completely respectfully and morally " (p8).

Based on students' experiences, their age, social status, and other background variables influenced their interpersonal interactions in e-learning. Also, soft or non-technical skills such as time management, teamwork, self-regulation, communication skills, and following the rules were reported as necessary from the student's perspectives. The majority of informants emphasized the number of learners in interpersonal communication. Some participants remarked on the knowledge and skills of learners about ICT.

" A very young person cannot establish a relationship with someone far from him in age. As a rule, there cannot be much empathy between these two people " (P.5).

" In groups, younger people or peers communicate better with each other " (P.13).

" In a virtual lesson or course, it is necessary that a person has some background or meta-cognitive skills " (P.4).

" The learner must strengthen these skills in herself/himself, such as time management, self-regulation, or metacognition skills; these can help her/him succeed in interaction " (P.4).

The majority of informants emphasized the standards of developing electronic content. They said that the teachers in multimedia need to follow the principles of voicing and reading from the slides in a monotonous tone. This issue needs better interaction between the teacher and the students. Some teachers even read from the text and do not feel like they are in the classroom as students.

" Sometimes, some lessons were essential, but our teacher spoke monotonously in multimedia. He either reads the text from that slide or lets someone else read it. You wanted to listen to your teacher's voice, but the voice was someone else's. You realized that the speaker of the text did not establish a connection with the subject and only read from it " (P7)

" Some teachers have only one voice in the audio files they upload. As if they are uploading a subject and reading it. You feel that it is not a classroom environment " (P3).

Participants emphasized logical sequence and coherency between materials.

" Teachers who go step by step are better than those who confuse many topics, and there is no connection between the topics. Teachers with continuity between their materials and go step by step are good " (P9).

In addition, the participants mentioned the applicability of the topic and their previous familiarity with it. In their opinion, this topic significantly impacted their relationship with the teacher.

" The student's degree of interest in the lesson is significant, that is, to feel how necessary and practical the lesson is " (P5).

" If the students had some background information about the subject the professor was teaching, their participation would be good " (P6).

## Theme 2: Features of the platforms

This category consists of asynchronous platforms, synchronous platforms, and social networks. Online learning can be organized into synchronous and asynchronous online courses. Asynchronous online learning does not occur in real-time but in the learner's program. It is on the idea of constructivist theory, a learner-centered approach that highlights the importance of peer-to-peer interactions. This approach incorporates self-study with asynchronous interactions to demand learning; it can also be applied to facilitate learning in traditional on-campus or traditional, distance, and continuing education. However, synchronous online learning occurs in real-time.

Some participants argued that the best platforms for asynchronous communication, stressing LMS, support Persian font (the common language of Iran). It is user-friendly, and it is possible to have permanent access to the content and track its general performance and advancement. It also has minimal technical issues.

" LMS must also be user-friendly " (P13).

" In the LMS we had, we had problems sending our assignments. Sometimes, the assignments went one day, and one day they did not; server and font problems were very difficult " (P13).

" In LMS, students' activities and interactions can be observed, and their progress can be tracked " (P4).

Synchronous learning occurs in real-time. Regarding the synchronous platforms, some participants emphasized the necessity of screen sharing. Some participants expressed concern about frequent outages due to poor internet connection and communication interference. On the contrary, they mentioned the possibility of grouping learners in small classes as an opportunity for some synchronous platforms.

" Adobe Connect was perfect, and it was possible to use all kinds of screens that supported the interaction of students with the teacher during teaching " (P13).

" The problem was the audio and image access; the sound was cut off and connected " (P3).

" I remember a good experience in the class of one of my teachers. The teacher came and formed several virtual rooms of three people and divided the students between the rooms, and each group had to discuss with each other. The teacher also entered the rooms and supervised " (P11).

The interesting subject that the participants highlighted was the opportunity of hearing the voice and seeing the image and body language of the teacher.

" Establishing emotional interaction happens only by hearing the teacher's voice and seeing the teacher's face " (P5).

" Seeing the teacher's face has a significant impact. Because you see the mimicry of his face, you see the movements created on his face and communicate. Regardless, you understand the message better " (P7).

Designing proper assignments was also vital, and many participants noted it.

" Homework should be something the student feels can be done for his benefit. Moreover, the assignments should be personalized " (p8).

" Students usually do their studies alone and do not interact with other students until they have to because of group activities " (p4).

Social media was created as a platform to share information. In this study, students emphasized their essential role and the need to strengthen social networks to improve interpersonal interactions in e-learning. They believed neural networks are a strong tool for reporting and sharing information.

" The messenger network provides the possibility of more interaction between the learners. According to the nature of the network, because it is much more accessible and manageable, it is possible to exchange messages faster in social networks " (p5).

" Social networks facilitate interaction. I think it was easier to interact through social networks " (p6).

## Theme 3: Systemic factors

This category consists of learner support and extracurricular activities.

Support is one of the critical elements of success for online students, which in the present study was also mentioned by students as a critical element to strengthen interpersonal communication.

One of the learners' expectations was educational support. Considering the registration and the nature of work and administrative affairs, which were all remote, they expected more guidance and support from the educational staff.

" The presence of the virtual education staff was weak. For example, maybe the education staff is only guided in choosing a unit " (P2).

Technical support was another form of support mentioned by participants. According to students, they should receive sufficient technical support for synchronous and asynchronous learning platforms.

" When LMS is used, the necessary support must be comprehensive " (p13).

" Even if they sell software, they support it 24 hours daily. Now, I am a poor student; sometimes I get confused, and I am looking for someone to guide me " (P10)

Also, Some participants remarked that extracurricular activities are necessary to strengthen interpersonal communication. According to the students, extracurricular activities will stimulate interaction.

" Scientific activities that stimulate interaction outside the classroom " (P6).

" One of the solutions is to create a space for them to interact outside the academic space, for example, an intellectual game, a simulation, or a game. Now, it can have an educational aspect, like gamification, or it can be just fun. They can be given an opportunity in that space, enter the game space as a team or any unique avatar, and interact with each other according to the goal of the game " (P16).

The design of scientific societies by students with the support of the university was also important.

" Now, the IR SOME Association (Iranian Society of Medical Educationalists) has been held. For the Science Olympiad, a number of students have registered, and the students themselves manage a sequence of workshops, so the interaction between the students is very good now. These groups were formed because all the students had a common goal, and the interaction was good " (P6).

" One of the solutions is to create a space for them to interact outside of the academic space " (P16).

## Discussion

Findings of this study demonstrated that students' experiences of e-learning fall into three themes: teaching and learning factors, system factors, and characteristics of platforms. The following factors affect teacher-learners and learner-learner interactions: instructor and learner characteristics, content characteristics and related factors, synchronous and asynchronous platforms, and learner support.

In terms of the first theme, factors related to teaching and learning, most students believed that there should be a positive and tension-free environment for learning, which is able to motivate learners to study hard. This finding is consistent with that of Xie and Derakhshan, who found that a positive teacher-student relationship could be established through empathy, caring, involvement, trust, and respect [34].Moreover, this result were consistent with data obtained in the study by Banks (2014). Banks have demonstrated that teachers should strive to create positive learning environments. Such teachers create caring environments; positive behavior is the focus of classroom support, and a change of direction instead of reprimands will be a tool to change behavior. Learners are offered a variety of choices to reach an agreed educational goal. Teachers who create positive environments pay close attention to all environmental stimuli [35]. Chen et al. reported that educators should be trained how to use social media to communicate online [36]. These results reflect those of Yu, who found that online instructors' factors, such as instructors' attitude to e-learning, knowledge, skill, and interaction with learners, affect learners' satisfaction with e-learning. Since, the interaction between

learners and instructors is the most important determining factor [37]. Their agility and adaptability are also skills needed to design online content [36]. Seifert demonstrated that learners expected their teachers to use technological tools appropriate to educational goals and content and to be aware of the benefits of technology [38]. Therefore, a teaching-learning process in the online environment requires more competencies than F2F, and it is true particularly in technology use [39].

This theme is in line with those of previous studies. The instructors' shift in pedagogy plays a critically important role in facilitating online interaction. In the e-learning environment, the instructor transmits knowledge through the relevant instructional design and technology. Therefore, the flexibility of the instructor, when presenting the content, supporting and communicating with students, and assessing them influences the students' evaluation of the course quality. Consequently, it affects their satisfaction with the whole learning experience. Such findings show that the quality of the instructor meets students' expectations. Instructors must design interactive teaching and facilitate continuous interactions to achieve student satisfaction [40]. In addition, the use of humor in teaching has been studied in recent decades in pedagogy. Researchers unequivocally note that humor enhances cognitive interest, stimulates positive feelings, and facilitates effective learning in students. Educator humor has the potential to encourage learners [41].

The present study clearly shows the significance of experience in terms of the way the teacher designs the assignments and the way he/she manages the class. Consistent with the findings of this study, siregar's qualitative study showed that students emphasized the significance of classroom management and that of designing activities aimed at enhancing interaction [10]. In this context, the participants all emphasized the importance of group tasks and their roles in improving interactions among learners. Siregar showed that the structure of assignments in the form of group or paired activities could increase interactions among learners [10].

The soft skills of learners were enumerated among the characteristics of learners. This agrees with Bergh et al.'s findings, which revealed that students had developed their soft skills and made their own identities through interaction with others, including teachers, peers, or patients [42]. Unlike hard skills that are among technical skills, soft skills are related to interpersonal skills, professional social skills, communication skills, and ethical attitudes [42]. As a result of their soft skills, moreover, students discussed abilities such as time management, communication, commitment to learning, and teamwork. This is consistent with the findings of Seifert (2021).In the mentioned study, the learners noted that instructor-student and student-student Interaction in an online course would increase learners' commitment to learning and success. They acknowledged that good time management for learning was critical for success in the course. They recommended that participants in online teaching should complete assignments immediately after receiving them [38].

The use of ICT facilitates effective engagement of the learners, enhances learning, and facilitates the use of teaching methods and materials to respond to students' interests and needs. In the present study, the following subcategories were extracted as novel findings: being busy, number of learners, and homogeneity of learners. It is necessary to consider these three factors for the effectiveness of facilitator interventions, which can include factors related to teaching and learning in order to increase the quality of learner-learner and learner-instructor interaction in their online courses.

To learners, the design of e-content in accordance with standards was also important. This is consistent with the findings of Alsadhan et al., who found that lack of standards is one of the factors affecting the development of multimedia e-learning systems [43]. One qualitative study pointed that the need for more skills for teachers in ICDL. According to that study, teachers

should become familiar with content development technologies. Therefore, the final result of their content production was non-standard [16].

Learner-content Interaction contributes predominately to the successful completion of the expected learning outcomes. To satisfy the learners, it is vitally for the e-learning content to include both excellent learning material and website content. As listening and reading alone cannot affect cognitive learning and generate knowledge, it is necessary to present these two skills through the interaction of students with the content, and this should be designed in a way to engage students in the online learning environment [44]. Thus, the instructors and administrators should pay attention to the content development and design of the course structure [44]. That is Edwards (2015) has argued that effective e-learning design should begin with what the learner needs to do rather than focus on what the learner needs to know. The quality of learning management system content for learners can be a predictor of good performance in e-learning environments and can lead to learner satisfaction [45].

As participants in a study have stated, it is possible to make communication more satisfactory and more comprehensible through adopting different integration patterns such as student-student and student-instructor dialogues, gestures, facial expressions, and tone of voice during e-learning [40], as stated by the participants in this study.

Feedback is how learners actively use comments and information from various sources to improve their learning [46, 47]. Although it can be time-consuming to provide or obtain feedback [48], it is crucial for an online instructor. In this study, the participants emphasized this issue with incredibly constructive, timely, and respectful feedback. They considered feedback as an essential way of communication with their teacher. Similarly, in qualitative research, participants' experiences demonstrated a lack of feedback. At the same time, they highlighted the constructive feedback of the teacher [10]. This is consistent with the findings of earlier studies. In a study, students stated that they expected the teacher to give them constructive feedback on their assignments during the course, not just at the end [38].

Another factor that influenced interpersonal interactions was the guidance of discussion forums by the teacher. All the participants wanted the active participation of the teacher, and this was mentioned by the learners in another study. They believed that when the discussion was going on in the forum, the learners liked to see the active participation of the instructor [38]. Discussion and learning networks provide opportunities to search, obtain, and present information [49], but the teacher must continuously follow up with the students.

Features of the platforms were the second theme. It is widely acknowledged that online networks are used as learning platforms that are widely distributed, more flexible, readily accessible, and, most importantly, permanently open. In one study, the effectiveness of the LMS was rated at 79% by learners reporting that they found it useful (89%), and that using it made their learning activities much easier (75.2%) [45]. If the system is user-friendly, the learner will use it frequently; therefore, better learning outcome achievements will occur, resulting in student satisfaction with e-learning. With excellent system quality, e-Learning occurs [40].

Learning media that are used in synchronous and asynchronous online learning, such as forum discussions, instant messaging, and blogs, play an important role in humanizing online courses by replicating the classroom experience of information exchange and social construct, not only between learners and instructors but also among the learners [50].

The primary function of LMS is to provide a safe and flexible learning platform for learners. It aims to organize, read, and maintain educational materials [51]. According to the participants, the user-friendliness of LMS had a significant impact on their interpersonal interactions.

Synchronous online learning happens in real time. This issue means that participants in this form of learning interact in a specific online environment at a given time [52]. Unfortunately, the participants' experiences of synchronous online platforms revolved around poor

internet network connectivity and interaction disruption. This was in line with the findings of previous studies. A study on medical students found that a poor internet connection was one of the most significant barriers to using online platforms [53]. Goli et al. reported that the poor internet bandwidth was an essential concern in the course of online teaching [54]. Participants also emphasized the critical role of social media in strengthening interpersonal interactions. According to Ofori et al., using social media for official academic communication creates an interactive learning environment, enhances social presence, and enhances learning outcome [55]. Social media platforms are one of the easiest and most effective means of disseminating information [56]. In this study, the participants repeatedly mentioned the advantage of social media in facilitating communication between the teacher and the students.

In the present study, students lauded the simultaneous use of synchronous and asynchronous learning. Due to the significance of these two types of learning, one study suggested the balanced combination of synchronous and asynchronous ways of learning [10].

Systemic factors and the necessity of student support were the final themes of this study. Cox applies the institutional theory to higher education institutions and concludes that six basic components underly the institutions' capacity to offer distance education courses. The six basic components are administrative commitment (allocating resources), online student support services (registration, advising, providing access), full-time online coordinator (assisting course development and online teaching materials), internal/external financial and technological resources (computers, online course management system), the online professional development (developing faculty online knowledge), and the adequate faculty participation (enough innovators supporting online education) [57].

Support is critical to online student success [47]. In the present study, participants were willing to receive educational and technical support and extracurricular activities. Learning support services include information, resources, personnel, and related interactive facilities. These resources cater to the diverse needs of students and effectively guide, assist, and promote their independent learning, collaborative learning, and holistic development [58]. Student support services must be integral to the education system to meet students' needs and expectations [59]. Zhang et al. showed that learning support services should focus on cognitive, emotional, and managerial aspects of online learning, meeting personalized learning needs, improving service quality, and promoting online learning [58], which was consistent with the findings of his study. However, in the present study, the main focus was on the improvement of interpersonal interactions in educational and technical aspects and extracurricular programs.

In the present study, the students, rather than teachers, were interviewed. It is therefore difficult, and time-consuming, to find a comprehensive picture of the phenomenon in question. However, the strength of this study is that it focuses on student-teacher interactions and student-student interactions, emphasizing different forms of synchronous and asynchronous communication. Furthermore, students at three universities were included in the study. In order to assess the diversity of the learners' views on different issues, it is suggested that more studies be conducted in other contexts.

## Conclusion

In general, the results indicated the importance of different aspects related to teaching-learning as facilitating factors of interpersonal interactions. Findings show that it is, crucial to develop a comfortable, dynamic, stress-free, and safe environment for the teaching-learning process to improve the interpersonal interactions. Educators should be provided with a series of pedagogical skills to support interactions. In addition, concentration and focus on some non-technical or soft skills of learners are also vital. Both educators and students should have achieved

adequate skills and knowledge in I.T. Learners need to strengthen soft skills such as time management and be able to take responsibility for their learning.

In addition to the significance of the teacher's leading role, the educational content must have critical standards, and it must be presented to the learners with a logical association. To strengthen interpersonal relationships, and in order to decrease the sense of social isolation, it is necessary to take into account the following variables: timely, respectful, and transparent feedback, the proper use of simultaneous and non-simultaneous communication tools and social networks. Furthermore, the presence of even short opportunities for F2F interaction and extracurricular activities can be sufficient for strengthening virtual interactions. Finally, comprehensive and continuous support of learners can improve the quality of interpersonal interactions.

## Implications

To strengthen the quality of interpersonal interactions (teacher-student and student-student interactions), empowering instructors and qualifying them is significant, considering their critical role in developing, maintaining, and facilitating the interaction of online classes. In addition, learners need to be trained to take responsibility for more participation in online classes and increase self-regulation skills and soft skills. In other words, the results of this study give teachers the insight to keep essential issues in mind when developing their online courses and students to be aware of their roles in the online learning process. Also, the characteristics of simultaneous and non-synchronous platforms, social messaging networks, and learner support are crucial.

## Supporting information

**S1 Checklist.**
(DOCX)

## Acknowledgments

The authors' gratitude goes to the participants who significantly contributed to this research.

## Author Contributions

**Conceptualization:** Rita Mojtahedzadeh, Shirin Hasanvand, Aeen Mohammadi.

**Data curation:** Shirin Hasanvand.

**Formal analysis:** Shirin Hasanvand.

**Investigation:** Rita Mojtahedzadeh, Shirin Hasanvand, Aeen Mohammadi.

**Methodology:** Rita Mojtahedzadeh, Shirin Hasanvand, Aeen Mohammadi, Sahar Malmir, Mehdi Vatankhah.

**Project administration:** Shirin Hasanvand.

**Supervision:** Rita Mojtahedzadeh, Aeen Mohammadi.

**Writing – original draft:** Rita Mojtahedzadeh, Shirin Hasanvand, Aeen Mohammadi, Sahar Malmir, Mehdi Vatankhah.

**Writing – review & editing:** Rita Mojtahedzadeh, Shirin Hasanvand, Aeen Mohammadi, Sahar Malmir, Mehdi Vatankhah.

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
