## [Decision Letter · Decision Letter 0]

14 Sep 2023

PONE-D-23-23929Students' Experience of Interpersonal Interactions Quality in E-Learning: A Qualitative ResearchPLOS ONE

Dear Dr. Hasanvand,

Thank you for submitting your manuscript to PLOS ONE. After careful consideration, we feel that it has merit but does not fully meet PLOS ONE’s publication criteria as it currently stands. Therefore, we invite you to submit a revised version of the manuscript that addresses the points raised during the review process.

Dear authors,Thank you for submitting your manuscript to PLOSE ONE.I have now received the reviews from the reviewers. After careful consideration, we feel that it has merit but needs revision. Please carefully address the comments and provide a line-by-line letter of response and highlight all the changes you make with different comments. If you disagree with the reviewers' comments, please write a rebuttal lefting your disagreement.Along with the reviewers' comments, please do consider the following points:1. The abstract is not informative. It lacks adequate knowledge about methodology. Moreover, whatever you wrote as Conclusion are not Conclusion but Implication. Please divide the Abstract into 6 parts: Background, Purpose, Method, Results, Conclusion, Implications2. The paper needs careful proofreading. It suffers from language problems, grammar, repetitions, tenses, subject-verb disagreement, etc)3. The introduction does not build a logical case and context for the problem statement.4. Theoretical framework/s and empirical evidence for introducing and supporting variables are weak in this paper.5. The introduction needs more focus on setting the context- describe the situation followed by the ‘problem’ which leads to the research question and your approach to solving the problem.More importantly, The paper lacks Statement of the Problem. What is (are) the problem(s) of this study? The authors should explain it in a separate subheading. In Statement of the Problem, the authors should first mention the problems and then bring at least one national finding and one international finding related to the problems.6. The Introduction should provide background information and the aims and rationale behind the paper. This will allow a clear understanding of the context and importance of the study. The “big picture’ relevance is important in the introduction which I couldn’t see it.7. The paper needs a literature review. In addition, and most importantly, the theoretical frameworks are missed from the manuscripts. For example, how does the data shape our current understanding of the topic? There was no conceptual framework to understand the data and the data was not as developed as I would expect to see in an academic journal article.8. To my humble opinion, discussion needs a deeper look.  In discussion: 1) the authors should mention the main reasons behind these results. Why such results obtained? What were the plausible reasons? 2) The authors should mention whether the result of their study is in line or in contrast with the previous studies by giving critical  reasons. 3) Discussion must be linked to theoretical frameworks. It can be strengthened by supporting the results with theoretical framework. It may mean that the authors must tell the readers and discuss what theory/theories support their findings. 9. Please add the implications of the study, limitations of the study, and suggestions for future researchers. 

10. Please add more updated references (2021-2023).Good LuckEhsan NamaziandostAcademic Editor

We look forward to receiving your revised manuscript.

Kind regards,

Ehsan Namaziandost

Academic Editor

PLOS ONE

Journal Requirements:

Reviewers' comments:

Reviewer's Responses to Questions

**Comments to the Author**

1. Is the manuscript technically sound, and do the data support the conclusions?

Reviewer #1: Yes

Reviewer #2: Yes

Reviewer #3: Yes

Reviewer #4: Partly

2. Has the statistical analysis been performed appropriately and rigorously? 

Reviewer #1: Yes

Reviewer #2: Yes

Reviewer #3: Yes

Reviewer #4: Yes

3. Have the authors made all data underlying the findings in their manuscript fully available?

Reviewer #1: Yes

Reviewer #2: Yes

Reviewer #3: Yes

Reviewer #4: No

4. Is the manuscript presented in an intelligible fashion and written in standard English?

Reviewer #1: Yes

Reviewer #2: Yes

Reviewer #3: Yes

Reviewer #4: No

5. Review Comments to the Author

Reviewer #1: The present study has been done with a clear method and the findings have been well defined and discussed. Tips for improving the article have been included in the text as comments, which are provided to you in the attached text

Reviewer #2: it seems that in the method of content analysis based on Granheim and Lundman's method, as explained in the method, codes and subcategories and category are presented, and categories are not combined and themes are not extracted.

In the presented table, it seems that the first column is not the subcategory, but the extracted codes, and the second column is the subcategories and the third column is the category.

system factors are mentioned and it is expected that it will have sub-layers as a layer. Based on the evidence of this study and personal experience, I would like to say that system factors can include two sub-categories, including student-related factors and teacher-related factors, and the codes related to each of these sub-categories can be educational support and support. technical support related to the student as well as educational support and technical support related to the teacher and according to experience and articles studied with the same analysis method, a category cannot end with one theme.

In the method, it s noted medical students, including doctors, dentists, etc., were interviewed, while only one nurse and midwife was interviewed, and the rest of the students were related to e-learning. Therefore, it is better to mention it as a limitation in generalizing the findings. Based on the results and limitations, suggestions for further studies should be provided.

Reviewer #3: A valuable manuscript on teaching and learning. The background and methods were explained well.

Abstract:

Consider using more specific terms when describing the research methods. For example, you mentioned "electronic interviews," which could be clarified as "online interviews" or "virtual interviews."

It might be beneficial to include a brief sentence highlighting the main practical implications of your study. What can educators, institutions, or policymakers learn from your findings.

please write e- learning in the keywords.

Introduction:

you've mentioned that "the quality of interpersonal interactions is a neglected link in e-learning," you could expand on this by briefly discussing why this gap in research exists and why it's important to fill it.

Carefully review the grammar, especially in longer sentences.

Methods:

Please provide a sufficient explanation about the field of study and students in the methods section.

Results:

In the result section, the third theme can emerge into the second one.

Conclusion:

It would be good if the author could explain the practical implication of this study.

Reviewer #4: The study suffers from some pitfalls. They are listed below:

- The introduction is chaotic. It does not mirror the significance of the study.

- The study lacks a rigorous literature review to explain the key concepts of the study.

- The design of the study is not clear. Explain why the authors have used this design.

- The participants section needs revisions to include more demographic information about the participants. Moreover, the author should explain how they accessed the participants and earned their consent.

- The procedures taken for data analysis should be expanded and verified.

- The data analysis procedures should be clearly mentioned and defined for readers

- Enough information about the results should be provided for readers.

- The qualitative results need to be revised. First present the theme, then define it, after that offer some excerpts to support the themes.

- The discussion section is really poor. It does not discuss the findings critically.

- The study does not have suggestions for further research.

- The study lacks implications for the pertinent stakeholders. Please fix this problem

- The references are not following APA guidelines. Please revise them.

With best regards,

6. PLOS authors have the option to publish the peer review history of their article (what does this mean?). If published, this will include your full peer review and any attached files.

Reviewer #1: **Yes: **Shadi Asadzandi

Reviewer #2: No

Reviewer #3: **Yes: **Mitra Amini

Reviewer #4: No

---

## [Author Response · Author response to Decision Letter 0]

27 Oct 2023

Along with the reviewers' comments, please do consider the following points: 

Thanks for your valuable comments

1. The abstract is not informative. It lacks adequate knowledge about methodology. Moreover, whatever you wrote as Conclusion are not Conclusion but Implication. Please divide the Abstract into 6 parts: Background, Purpose, Method, Results, Conclusion, Implications. It was corrected (P: 2,3)

2. The paper needs careful proofreading. It suffers from language problems, grammar, repetitions, tenses, subject-verb disagreement, etc). The whole of the paper was reworked (P:4-7).

3. The introduction does not build a logical case and context for the problem statement. It was corrected(P:4-7).

4. Theoretical framework/s and empirical evidence for introducing and supporting variables are weak in this paper. The empirical evidence for supporting variables was added(P:4-7).

5. The introduction needs more focus on setting the context- describe the situation followed by the ‘problem’ which leads to the research question and your approach to solving the problem. More importantly, The paper lacks Statement of the Problem. What is (are) the problem(s) of this study? The authors should explain it in a separate subheading. In Statement of the Problem, the authors should first mention the problems and then bring at least one national finding and one international finding related to the problems. The Introduction should provide background information and the aims and rationale behind the paper. This will allow a clear understanding of the context and importance of the study. The “big picture’ relevance is important in the introduction which I couldn’t see it. The statement of the problem was corrected(P:4-7).

6. The paper needs a literature review. In addition, and most importantly, the theoretical frameworks are missed from the manuscripts. For example, how does the data shape our current understanding of the topic? There was no conceptual framework to understand the data and the data was not as developed as I would expect to see in an academic journal article. It was corrected(P:4-7).

7. To my humble opinion, discussion needs a deeper look. In discussion: 1) the authors should mention the main reasons behind these results. Why such results obtained? What were the plausible reasons? 2) The authors should mention whether the result of their study is in line or in contrast with the previous studies by giving critical reasons. 3) Discussion must be linked to theoretical frameworks. It can be strengthened by supporting the results with theoretical framework. It may mean that the authors must tell the readers and discuss what theory/theories support their findings. The discussion was revised and modified again(P:16-21).

8. Please add the implications of the study, limitations of the study, and suggestions for future researchers. They were mentioned in the discussion(P:22).

9. 10. Please add more updated references (2021-2023). Updated references were used and modified(P:26-8).

---

## [Decision Letter · Decision Letter 1]

29 Nov 2023

PONE-D-23-23929R1Students' Experience of Interpersonal Interactions Quality in e-Learning: A Qualitative ResearchPLOS ONE

Dear Dr. Hasanvand,

Thank you for submitting your manuscript to PLOS ONE. After careful consideration, we feel that it has merit but does not fully meet PLOS ONE’s publication criteria as it currently stands. Therefore, we invite you to submit a revised version of the manuscript that addresses the points raised during the review process.

We look forward to receiving your revised manuscript.

Kind regards,

Ehsan Namaziandost

Academic Editor

PLOS ONE

**Additional Editor Comments:**

Dear authors,

Thank you for the revision. Unfortunately, the reviewers, as well as I, are still not satisfied with the revision and raised serous points. The reviewers claimed that if their comments are not considered very carefully, the manuscript will be rejected.

Thus, I would like to give you another chance to revise to your paper based on the reviewers comments. Please highlight the changes and provide a point by point response to the comments.

All the best,

Ehsan Namaziandost

PLOS ONE Editor

Reviewers' comments:

Reviewer's Responses to Questions

**Comments to the Author**

1. If the authors have adequately addressed your comments raised in a previous round of review and you feel that this manuscript is now acceptable for publication, you may indicate that here to bypass the “Comments to the Author” section, enter your conflict of interest statement in the “Confidential to Editor” section, and submit your "Accept" recommendation.

Reviewer #1: All comments have been addressed

Reviewer #2: All comments have been addressed

Reviewer #3: All comments have been addressed

Reviewer #4: (No Response)

Reviewer #5: All comments have been addressed

2. Is the manuscript technically sound, and do the data support the conclusions?

Reviewer #1: Yes

Reviewer #2: Yes

Reviewer #3: Yes

Reviewer #4: Partly

Reviewer #5: Yes

3. Has the statistical analysis been performed appropriately and rigorously? 

Reviewer #1: Yes

Reviewer #2: N/A

Reviewer #3: Yes

Reviewer #4: No

Reviewer #5: Yes

4. Have the authors made all data underlying the findings in their manuscript fully available?

Reviewer #1: Yes

Reviewer #2: Yes

Reviewer #3: Yes

Reviewer #4: Yes

Reviewer #5: Yes

5. Is the manuscript presented in an intelligible fashion and written in standard English?

Reviewer #1: Yes

Reviewer #2: Yes

Reviewer #3: Yes

Reviewer #4: No

Reviewer #5: Yes

6. Review Comments to the Author

Reviewer #1: All my comments have been applied in the article and I think the article has no problem and has been accepted

Reviewer #2: (No Response)

Reviewer #3: Dear authors

Thanks for revision. The manuscript is now acceptable. You answered all of my comments.

Reviewer #4: Though the authors made some modifications, there are still major issues with the manuscript. I just listed some of them below:

- There are many claims in the introduction part which need citations. Please fix this problem.

- The citations are not following the format of the journal. Please fix this problem.

- The significance of the study has not been well discussed in the introduction parts. It is not clear why the authors decided to conduct this study.

- The theoretical background of the study is missing. The authors are supposed to explain the theoretical underpinning on which the study has been rested.

- The required information about the participants should be discussed. Moreover, the authors are supposed to explain how they met the ethical requirements during the study.

- The data collection procedures are quiet vague. Please rephrase this part.

- The procedures taken to analyse the gained data are quiet problematic. The authors should explain in details the ways through which they might have analyzed the data. Additionally, it should be reported the ways through which the authors measured the reliability and validity of the gained findings.

- For the findings sections, follow the following blueprint:

Theme …. Definition of Them …. Excerpts support the theme

- The discussion part needs to be critical. The authors need to revise it to make it more critical based on the available literature.

- Given the limitations imposed on the study, some suggestions for further research need to be offered for potential readers.

- The references are not following the APA 7th style.

- The language of the manuscript needs substantial improvement. There are grammatical and lexical problems throughout the manuscript.

With best regards,

Reviewer #5: Dear authors,

I had the chance to review the revised version of the paper "Students' Experience of Interpersonal Interactions Quality in e-Learning: A Qualitative Research". I provided a detailed review/evaluation referring to each comment of the previous reviewers with the correction made by the authors in the manuscript. In my overall opinion, the revised version has met the standards of the PLOS ONE journal and could be considered for publication.

Best regards,

7. PLOS authors have the option to publish the peer review history of their article (what does this mean?). If published, this will include your full peer review and any attached files.

Reviewer #1: No

Reviewer #2: No

Reviewer #3: **Yes: **Mitra Amini

Reviewer #4: **Yes: **Afsheen Rezai

Reviewer #5: No

---

## [Author Response · Author response to Decision Letter 1]

5 Jan 2024

Review Comments to the Author

Reviewer #1: All my comments have been applied in the article and I think the article has no problem and has been accepted

Reviewer #2: (No Response)

Reviewer #3: Dear authors

Thanks for revision. The manuscript is now acceptable. You answered all of my comments.

Reviewer #4: Though the authors made some modifications, there are still major issues with the manuscript. I just listed some of them below:

- There are many claims in the introduction part which need citations. Please fix this problem. All mentioned phrases are cited in the text of the introduction. Again, the background and review of the literature were revised.

- The citations are not following the format of the journal. Please fix this problem. They were corrected.

- The significance of the study has not been well discussed in the introduction parts. It is not clear why the authors decided to conduct this study. Done 

- The theoretical background of the study is missing. The authors are supposed to explain the theoretical underpinning on which the study has been rested. Done

- The required information about the participants should be discussed. It has been mentioned. (P:9)

Moreover, the authors are supposed to explain how they met the ethical requirements during the study. It has been mentioned. (P:9,24)

- The data collection procedures are quiet vague. Please rephrase this part. It was corrected.

- The procedures taken to analyses the gained data are quiet problematic. The authors should explain in details the ways through which they might have analyzed the data. Thanks for your valuable comment. In the statistical analysis section, we talked about the data analysis method in detail. At first, the data analysis method and its steps are described. In the following, we have said what we have done at each step. We added some explanations again. I hope it is clear and clears our confusion. The method of data analysis is well-known in health care research and is frequently used by researchers in qualitative content analysis. The reference is also introduced. We used Graneheim and Lundman (2004) method. 

Additionally, it should be reported the ways through which the authors measured the reliability and validity of the gained findings. Thanks for your comments. The validity and reliability of the data are mentioned with its equivalent term in qualitative research called Trustworthiness and based on the 4 criteria of Lincoln and Guba in the methodology section. Coding was done only by one of the authors, and there was no need to calculate the agreement between the coders or the intercoder rater.

- For the finding’s sections, follow the following blueprint:

Theme …. Definition of Them …. Excerpts support the theme: According to your valuable comment, they were corrected.

- The discussion part needs to be critical. The authors need to revise it to make it more critical based on the available literature. It was corrected.

- Given the limitations imposed on the study, some suggestions for further research need to be offered for potential readers. It was mentioned and highlighted (P:23)

- The references are not following the APA 7th style. PLOS uses “Vancouver” style, as outlined in the ICMJE sample references. ( https://journals.plos.org/plosone/s/submission-guidelines)

- The language of the manuscript needs substantial improvement. There are grammatical and lexical problems throughout the manuscript. It was corrected. It was corrected. All phrases and words in the text of the paper are marked in red.

---

## [Decision Letter · Decision Letter 2]

18 Jan 2024

Students' Experience of Interpersonal Interactions Quality in e-Learning: A Qualitative Research

PONE-D-23-23929R2

Dear Dr. Hasanvand,

We’re pleased to inform you that your manuscript has been judged scientifically suitable for publication and will be formally accepted for publication once it meets all outstanding technical requirements.

Kind regards,

Ehsan Namaziandost

Academic Editor

PLOS ONE

Additional Editor Comments (optional):

Reviewers' comments:

Reviewer's Responses to Questions

**Comments to the Author**

1. If the authors have adequately addressed your comments raised in a previous round of review and you feel that this manuscript is now acceptable for publication, you may indicate that here to bypass the “Comments to the Author” section, enter your conflict of interest statement in the “Confidential to Editor” section, and submit your "Accept" recommendation.

Reviewer #1: All comments have been addressed

Reviewer #2: All comments have been addressed

Reviewer #4: All comments have been addressed

Reviewer #5: All comments have been addressed

2. Is the manuscript technically sound, and do the data support the conclusions?

Reviewer #1: Yes

Reviewer #2: (No Response)

Reviewer #4: Yes

Reviewer #5: Yes

3. Has the statistical analysis been performed appropriately and rigorously? 

Reviewer #1: Yes

Reviewer #2: (No Response)

Reviewer #4: Yes

Reviewer #5: Yes

4. Have the authors made all data underlying the findings in their manuscript fully available?

Reviewer #1: Yes

Reviewer #2: Yes

Reviewer #4: Yes

Reviewer #5: Yes

5. Is the manuscript presented in an intelligible fashion and written in standard English?

Reviewer #1: Yes

Reviewer #2: Yes

Reviewer #4: No

Reviewer #5: Yes

6. Review Comments to the Author

Reviewer #1: Hello

All the comments have been well applied by the authors and in my opinion there is no problem in the work and the work is approved

Reviewer #2: (No Response)

Reviewer #4: Before sending the manuscript to publication office, the manuscript should be proofread one more time. At the present status, it is not readable enough.

Reviewer #5: Dear author,

Thank you for considering my comments on your paper. In my opinion the revised version has met the standards of the PLOS ONE journal and could be considered for publication.

Best regards,

7. PLOS authors have the option to publish the peer review history of their article (what does this mean?). If published, this will include your full peer review and any attached files.

Reviewer #1: No

Reviewer #2: No

Reviewer #4: No

Reviewer #5: No

---

## [Editor Report · Acceptance letter]

16 Mar 2024

PONE-D-23-23929R2 

PLOS ONE

Dear Dr. Hasanvand, 

I'm pleased to inform you that your manuscript has been deemed suitable for publication in PLOS ONE. Congratulations! Your manuscript is now being handed over to our production team.

Kind regards, 

on behalf of

Dr. Ehsan Namaziandost 

Academic Editor

PLOS ONE